# Emerging Perspectives on Non-Chemical Weed Management Tactics in Container Ornamental Production in the United States

Greeshmanth Alluri and Debalina Saha *

Department of Horticulture, Michigan State University, 1066 Bogue Street, East Lansing, MI 48824, USA;
allurigr@msu.edu
* Correspondence: sahadeb2@msu.edu; Tel.: +1-517-353-0338

**Abstract:** Weeds are undesirable plants that can interfere with human activities and can hamper crop production and practices. The competition among ornamentals and weeds for space, nutrition, light, and moisture within a restricted area, such as in container production, can be intense and destructive. In response to increasing concerns regarding herbicide injuries and the effects of pesticide use on the environment, many growers are extremely interested in non-chemical pest-management approaches. There are various non-chemical strategies to control weeds in containers, which include scouting, sanitation practices, hand weeding, mulching, irrigation management, substrate stratification, mulch discs or geo discs, lid bags, and fertilizer placement. In a restricted growth environment, weeds have been shown to reduce crop growth significantly. Limited information is available on the effects of weed densities and container sizes on ornament–weed competition within containerized production and how the concepts of fertilizer placement can be used efficiently to control weeds in containers without using any herbicides on the ornamentals. There is an immediate need to evaluate the interference and competitive effects of pernicious weed species in container-grown ornamentals in the North Central United States and to develop effective non-chemical weed control strategies by altering fertilizer placement in container production.

**Keywords:** weed control; nursery production; non-chemical; ornamental crops





## 1. Introduction

Controlling weeds in nursery container production is an important aspect, as they can compete with ornamental plants for soil, nutrients, water, light, and space within the container. As a result, there is a decrease in the quality, aesthetic, and market value of the ornamental plants, and sometimes ornamental crops can die due to severe competition with heavy infestations of weed species. In addition, weeds can harbor insects, pests, diseases, and pathogens, resulting in further reduction of market value.

Other than a select group of graminicides, which can be applied to certain ornamentals, there are virtually no post weed control options in container nursery production other than hand weeding. Thus, weed control is typically achieved through the use of pre-emergence herbicides in combination with supplemental hand weeding. Weed control in container nursery production is often the highest production cost encountered by nursery growers, often exceeding USD 4000 per acre [1,2]. A recent study in 2017, conducted by Ingram et al. [3], on three production scenarios for *Buxus microphylla* var. japonica 'Green Beauty' showed that weed control (which included hand weeding and herbicide application costs) is an important component for variable costs in nursery production as it can cost 38% of the total production cost per shrub in field production and 7% of the total production cost per shrub in number 1 containers [3]. In Michigan, the total financial impact of nursery and landscape production, including backward linked industries, is USD 1.26 billion [4]. According to Kundson [5], the ornamental/floriculture industries of Michigan directly employ 13,269 people and a total of 16,663 employees. So, a little

improvement in weed control can help these nursery growers and greenhouse operators to improve their overall profitability and thereby directly impact Michigan's billion-dollar nursery production. If weeds are successfully controlled there can be improvements in the overall quality and market value of ornamentals and a reduction in labor costs, thereby increasing the profit margins of Michigan growers.

Over the past twenty years, ornamental production with respect to the world has changed significantly. In the global scenario of ornamental and cut flowerpot plant sales, the U.S. accounts for 12.5% in total and stands amongst the top three nations, preceded by China (18.6%) and Europe (31.0%) with second and first places, respectively. The greatest challenge for ornamental production is sustainability, and data to highlight the problematic concerns which need improvement through environmental contributions pertaining to production, storage, and transportation can be obtained from life cycle assessments (LCAs) of ornamentals [6].

In the U.S. for the past 15 years, amongst the nursery industry segment container production is a rapidly growing sector, which is likely to expand further. The critical problem faced in container-grown nurseries is weed infestations because resources like water, nutrients, and the availability of soil–air are restricted to the capacity of the container, for which they compete with the main plant or ornamental [1]. According to growers in nurseries, based on the type of weed species being removed from the containers manually, they allocate an estimated cost of USD 500 to USD 4000/acre or USD 1235 to USD 9880/ha. A value of approximately USD 7000/acre or USD 17,290/ha is the estimated economic damage occurring due to the infestation of weeds [2]. The efficient control of weeds is indeed crucial because the intensity of destruction they create is undervalued generally [1]. If weed control approaches are upgraded by reducing the cost of expenses, it would remarkably impact the industry in a positive way [2].

To manage weeds efficiently, a good aggregate of robust sanitary measures, best cultural practices, and proper usage of pre-emergence herbicides is required. In container production, preventive measures are important to control weeds because in field production, crops are sown directly into the soil which makes it easy to effectively control weeds even after their germination by applying herbicides or by mechanical cultivation, whereas in the case of container crops a direct spraying of herbicides is not appropriate. So, for profitable weed management in containers it is always suggested to prevent weeds even before they germinate [7]. Chemical methods of weed control may not be applied to greenhouse/enclosed structures or to several sensitive ornamental species due to the potentiality of herbicide injury. In addition, over usage of herbicides can lead to environment-related issues such as surface runoff, ground water contamination, and the off-target movement of herbicides. Repeated application of the same herbicide with a similar mechanism of action can even cause herbicide resistance among weed species. Taking these into consideration, non-chemical methods are more environmentally friendly in these instances. However, growers may not be aware of different types of non-chemical techniques that can be applied to these ornamental production systems. Hence, the objectives of this literature review article are to summarize the previous and current non-chemical weed control practices that are prevalent in the United States's ornamental crop production systems and to discuss the knowledge gaps and future research directions.

## 2. Importance of Ornamental Crop Production

The Green Industry, also referred to as the U.S. environmental horticulture sector, consists of various entities, such as horticultural nurseries and turf producers, contractors, landscape designers, retail garden centers, maintenance firms, home centers, mass merchandisers with landscape departments, and middlemen like brokers and horticultural distribution centers. Regardless of economic downturns, this industry is among the fastest-growing sectors within the country's agricultural economy, and it is frequently expanding and advancing [8]. Over the past three decades, the Green Industry has been known for its rapid growth, innovation, and constant evolution. Nonetheless, as demand growth slows

down and operating margins become compact, it suggests that the industry is entering a stage of maturation [9]. In 2018, a survey (Green Industry Research Consortium) was conducted among 1727 participants to report their annual sales, which amounted to a total of USD 2.392 billion (B). On average, each firm had sales of USD 1.39 million (M). Wholesale market channels had sales totaling USD 1.74 B and averaging USD 1.34 M per firm, while retail sales had a total of USD 474 M, with an average of USD 0.53 M per firm. The survey also showed that the Southeast region stood out with annual sales of USD 542 M, while the Midwest (USD 489 M) followed by the Pacific (USD 485 M), Southcentral (USD 280 M), Northeast (USD 276 M), Appalachian (USD 135 M), Mountain (USD 125 M), and Great Plains (USD 61 M) reported lower sales. The retail sales accounted for 20 percent of the overall reported annual sales and varied across the regions, ranging from 7 percent in the Southeast to 73 percent in the Great Plains [10].

In 2019, the wholesale value of floriculture crops decreased by 7% when compared to the previous year. The top five states in 2019 were California, Florida, Michigan, New Jersey, and Ohio, which accounted for 69% of the total value or USD 3.04 billion [11].

In 2020, the wholesale value of floriculture crops increased by 9% when compared to the previous year. Growers with USD 10,000 or more in sales are estimated to have contributed to a total crop value of USD 4.80 billion, which is higher than the USD 4.42 billion recorded in 2019. The top five states in 2020, including Florida, California, Michigan, New Jersey, and Ohio, were responsible for 65% of the total value, with a combined contribution of USD 3.13 billion [12].

In 2018, the U.S. Green Industry survey respondents had a workforce of 35,719 individuals, with permanent employees making up 57.8% (20,631); temporary, part-time, or seasonal employees accounting for 35.4% (12,633); and out of the total number of employees, 6.9% (2455) were foreign national workers who are permitted to work in the United States under the H2A visa program. The regions with the most reported employment were the Southeast and Midwest, with 10,474 and 9162 employees, respectively [10].

The mean number of employees per company across the country was 20.8, comprising 11 full-time and permanent staff; 7.5 part-time, temporary, or seasonal workers; and 1.9 H2A laborers [10].

However, many firms (71%) indicated that they maintained their number of full-time/permanent employees over the past five years. Meanwhile, 11% of firms reported a decrease in employment, and 19% reported an increase. Similarly, around 68% of firms kept their number of part-time/temporary/seasonal employees consistent, while 12% reduced employment and 20% increased it [10].

## 3. Impact of Weeds on Ornamental Crop Production

Weeds are unwanted and undesirable plants that can hamper crop production and practices and can interfere with human activities. In a restricted area such as in container production, the competition between ornamentals and weeds can be intense and destructive as weeds compete for space, nutrient, moisture, and light. They cause substantial environmental damage and are accountable for ample losses [13] in ornamental production. In container-grown ornamentals, weed competition can greatly decrease the shoot dry weight of intended plants; for example, one eclipta per pot would reduce shoot growth by 43% on 'Fashion' azalea shoots [14]. Researchers have demonstrated that depending on the weed species, even one weed in a tiny (3.78 L) pot can affect the growth of an ornamental crop [1].

According to Berchielli-Robertson [14], the level of competition between weeds and ornamental plants varies, because in woody plants development during container production is greatly inhibited by competition from some weeds, but not all. Although, if weeds did not limit growth, a weed-infested container plant is a less marketable product than a weed-free product [1]. This is because the aesthetic value of the ornament reduces with the weed infestation, which becomes less attractive to the customers. According to Khamare et al. [15], in a nursery environment, there were reductions in biomass regardless of the species or container size. Furthermore, even when the plant growth indicators were

comparable, the presence of weeds caused significant production delays. In addition, as the weed density rises, competition effects may begin slowly at first and then become more severe. After a certain point, no more effects would be seen as the plants experience heavy weed pressure. For example, when Japanese holly (*Ilex crenata* Thunb.) and ligustrum (*Ligustrum vulgare* L.) are grown with different levels of weeds in containers of different sizes, the shoot dry weight of Japanese holly was evidently reduced by 18% and 22%, 51% and 52%, 51% and 53%, and 40% and 53% in 3.8 L, 11.4 L, 24.7 L, and 56.8 L containers, respectively. On the other hand, the shoot dry weight of ligustrum was observed to be reduced by 28% and 35%, 55% and 56%, 41% and 43%, and 12% and 14% in same-sized containers, respectively [15].

Restricting annual weed growth in container production has turned into a major economic concern for growers [16]. While there has not been much research on managing weeds in containers, growers frequently use a variety of weed management techniques in this area [17]. In the current scenario, various chemical and non-chemical methods of weed control are in existence [2]. The use of chemical weed control in container nurseries has become a norm since the 1970s, when it was estimated that weed management accounted for approximately 20% of the overall production cost [17]. The key to successful weed control with herbicides involved a three-stage approach. Firstly, if possible, weeds should be eradicated before planting by using a comprehensive post-emergence herbicide or soil sterilant. Annual weeds can be removed through cultivation; however, herbicides are more effective in eliminating perennials or weeds that have developed in underground storage tissue. Secondly, the emergence of new weeds should be deterred by applying pre-emergence herbicides, which is the primary method of controlling weeds in nurseries. Lastly, any escaped weeds should be tackled using post-emergence weed control techniques. However, it is not possible to find a single herbicide that can manage all weed types [18]. Oxyfluorfen, isoxaben, and simazine are pre-emergence herbicides that are effective against weeds of broadleaves. On the other hand, prodiamine, pendimethalin, and oryzalin are useful for controlling grasses and few small-seeded broadleaf weeds of pre-emergence, whereas Fluazifop-butyl, clethodim, and sethoxydim are post-emergence herbicides that are specifically designed for controlling grass weeds. For broad-spectrum weed control through directed spray applications, nonselective post-emergence herbicides like glufosinate, paraquat, and glyphosate are employed [18].

Going further, spreading granular herbicides with a cyclone spreader over the top of stock is a prevalent method for controlling weeds in containers. But, at the same time applying three to five granular herbicides a year resulted in consequent non-target herbicide damage [2]. According to Carpenter [19], nursery stock grown in containers with the highly porous media can be infused with activated carbon (C), upon which it is safe to incorporate broad-spectrum herbicides onto the surface of the container's activated C-free layer. So, injuries are prevented because the herbicides appear to become absorbed by the activated C before the plant roots do. However, herbicides have always had moderate success in reducing labor. A broad-spectrum herbicide, such as dichlobenil, provides significant weed control, but crop damage is possible [19].

In response to increasing concerns regarding the effects of pesticide use on the environment, many growers are extremely interested in non-chemical pest-management approaches [20]. There are various non-chemical strategies to control weeds, which include prevention and exclusion, hand weeding, using mulch and cover crops, heat, weed mats, geo discs, and organic products. Although some of these options may only be suitable for weed management in containers, all of them can be used around containers and in non-crop areas. Additionally, it is rare to rely on just one alternative method as they tend to be less effective individually than synthetic herbicides. To achieve optimal weed control, it is often necessary to employ a combination of two or more alternatives [21]. For quite some time, cultural weed control methods like mulching have been utilized in outdoor spaces to prevent weed growth, yet their ability to control weeds in container plant production has not been fully explored. Another method that may help manage weeds is subirrigation,

as long as the top layer of the potting mix remains dry, which makes it a less suitable environment for weed seeds to germinate and establish themselves [22]. The cost of manual hand weeding can be substantial, but it can be a suitable option for a small nursery environment. It is important to address weed growth early on when they are still small, as removing larger weeds from containers can result in a significant loss of growing media [23]. An important initial step in mitigating weed growth is to implement appropriate sanitary procedures during the production of liners and propagation. The "Sanitation-Exclusion-Prevention" approach helps to diminish or eliminate the growth and spread of weed seeds and propagules, making weed control efforts more manageable. Even simple measures such as cleaning equipment and containers and covering the storage areas of substrates can have a substantial impact on reducing weed prevalence [20]. According to Diver et al. [23], a new weed control agent, corn gluten meal, has been recently introduced into the market. It is derived from the processing of corn syrup and serves as a bioherbicide. During the early spring, corn gluten meal is applied as a pre-emergent herbicide and its effectiveness is optimized when spread over the top one-fourth inch of soil. However, annual reapplication is necessary to maintain its potency. This meal comprised 10% nitrogen and provides a gradual nutrient release to the crops as a slow-release fertilizer. It has been patented and commercially sold as an herbicide [23]. But newer studies have uncovered that corn gluten hydrolysate (CGH), produced from corn gluten meal, outperforms corn gluten meal in weed management for cut flowers and can be applied at a lower rate for effective results [24]. Weed management practices are extremely site-specific and significantly different from one region to another; the nursery industry faces numerous challenges for the development of this knowledge [25].

## 4. Non-Chemical Techniques for Weed Control in Container Nursery Production

The following subsections are some of the major non-chemical techniques used for weed control in container nursery productions:

### 4.1. Scouting

Weed identification and thorough scouting are necessary for efficient weed control. The most important thing to keep in mind is that control should be exercised rather than total eradication [1]. Weed scouting plays a crucial role in contemporary integrated weed management; however, when performed manually, it could be laborious and time consuming [26]. Once identified, weeds should be categorized according to their lifespan, with perennials being more difficult to control. Weeds that have resisted current weed management methods and those listed as noxious by state or federal authorities must be given utmost priority. Additionally, any new weed species found should receive special consideration [1].

Historically, weed information has been gathered casually and without much regard for the species of weeds, their distributions, or densities. This was primarily due to the time and labor required to conduct thorough scouting, the resulting information being complicated, and the assumption that weeds were consistently and evenly distributed across a field. Furthermore, even if variations were detected, there was insufficient equipment to address them [27]. According to Wiles et al. [28], to recommend a post-emergence control treatment, the weed seedlings in a field must be sampled or scouted to determine the most appropriate treatment. Additionally, the effectiveness of the scouting approach must be evaluated to ensure that it is a cost-effective solution. So, to select an effective post-emergence weed control strategy, the dominant weed species needs to be identified. To achieve this, a scouting plan must be developed that specifies the shape and size of the quadrata, or sample units, in which weeds will be identified and counted. The sampling intensity, or the number of quadrats to be examined, is also included in the plan, as is the sampling strategy for determining the location of the quadrats within the field. This ensures a distinct approach to weed management [28].

It is recommended to conduct container nursery weed monitoring at least three to four times annually. The initial assessment should take place in the spring, with the aim of identifying weeds that managed to evade the fall pre-emergence program, as well as winter annuals that are currently sprouting. This should be followed by one or more summer evaluations to locate summer annuals that slipped through the spring pre-emergence program, as well as winter annuals that are persisting. Lastly, before the first frost in the fall, it is important to spot summer annuals and perennials that were not successfully controlled, as well as winter annual seedlings [1].

Weed scouting is typically performed by manually inspecting a field and using sampling techniques to estimate weed species distribution. This process can be time-consuming, making it an ideal candidate for automation. By utilizing robots to scout the entire field, humans could make decisions about weed management based on the robot's findings. Despite recent efforts to develop robots for automated weed control and scouting, critical areas still require improvement before these systems can be widely adopted [26].

### 4.2. Sanitation Practices

To effectively manage pests or weeds in a nursery or greenhouse setting, the most important factor is prevention. This can be accomplished by prioritizing sanitation practices to reduce the introduction and spread of weeds, insect pests, and diseases in greenhouse and nursery environments. It is crucial to use well-maintained tools and equipment and consistently adhere to sanitation protocols to prevent pests from being transferred through these channels. Weeds or pests can be contained by limiting the movement of non-sterile equipment, vehicles, and individuals around the setting. However, it is also crucial to set up a framework for comprehensive sanitation management and give training to guarantee that staff members adhere to correct sanitation procedures, which ensures a complete sanitation management plan [29].

The use of greenhouses for growing plants allows growers to prevent or minimize the entry of weed seeds. Nevertheless, weed seeds may still find their way into greenhouses through openings such as vents, windows, or doors. They also have the potential to be transmitted through water or introduced through plant materials, tools, equipment, human interventions, or animals. Ensuring that all the pavements and aisles leading to the greenhouse entrance are clear of vegetation, or mowing any grass and other vegetation on a regular basis and keeping them close to the ground will help to prevent weed seeds from being carried in by foot traffic. To further decrease the number of wind-borne seeds entering the greenhouse, consider using screen exclusions on the vents or windows. It is also important to keep the areas beneath the benches free of container media and plant debris as this will reduce weed germination. To prevent further weed seed germination and facilitate easy cleanup, one can consider using concrete floors or weed barrier fabrics over gravel. If intending to reuse containers, they should be washed thoroughly using pressurized water flow and chemical disinfectants to eliminate any dirt, pathogens, and weed seeds [30].

However, identifying the source of weed seeds in any nursery can be more challenging than it appears. Based upon the circumstances prevailing in each nursery, several sources for these seeds can differ. Some possible sources may include potting substrates, nearby areas where weeds are growing, and sometimes the pots themselves. Weed seeds are usually not present in potting materials such as pine bark, peat moss, and perlite. Even so, weeds may infiltrate these substrates if they are stored in bulk, either at the nursery premises or at the substrate supplier's location. Weeds in the proximity of production beds or substrate piles can introduce weed seeds through various means, including wind, physical dissemination {for example, bittercress (*Cardamine* sp.), which can disperse its seeds over several meters away}, and invasion by certain weeds that possess stoloniferous and rhizomatous traits [31].

During the period when the beds are not occupied by crops, it is critical to take steps to eliminate any existing weeds, either physically or chemically. If necessary, one can

replace the old weed fabric or stones with new ones and sweep away any existing debris. To summarize, the simplest and most effective way to reduce weed seeds in containers is to practice good sanitation, specifically by keeping non-crop areas weed-free [7].

### 4.3. Hand Weeding

Hand weeding may be the most opted-for or recommended method of controlling weeds in ornamental production sites or nurseries where they are dispersed. Hand weeding, while time-consuming, should be an essential component of any weed management program to prevent weeds from reproducing or seeding. Consistent weed removal while they are small and before they start seeding can significantly reduce the number of annual weeds over time. It is also suggested that hand weeding must be conducted on a regular basis until plantings become well established [32].

Hand weeding necessarily requires significant amounts of both labor and financial resources, additional to the expense of herbicides [33]. The production of ornamental plants in container culture remains a highly challenging task, primarily due to the lower availability of post-emergence herbicide options and herbicide-sensitive ornamental plants, which leads to heavy dependence on hand weeding. Additionally, the wide diversity of crop species further exacerbates the difficulties of weed control in container production [25]. Further, growers of minor crops require more efficient herbicides as well as affordable or cost-effective alternatives to hand weeding to reduce the expenses associated with weed control [34].

Depending on the size of the nursery, annual weeding labor expenses varied from USD 608 to USD 1401 per hectare (equivalent to USD 246–USD 567 per acre). Nurseries with a land area of 4.4–9.7 hectares (11–50 acres) had lower costs, while those with a land area of less than 4 hectares (10 acres) and more than 20.2 hectares (50+ acres) faced higher expenses. Hourly wages in various nursery sizes were comparable, falling within the range of USD 3.53 to USD 3.97 [17]. According to North Carolina reports, the cost of hand weeding 1000 pots over a four-month period could be USD 1367 if no herbicides are used. This estimate is based on an hourly wage of USD 14.75, which is typically paid for labor by local nurseries [35].

The cost of production will rise because of higher labor expenses unless other instruments, like new herbicides and precision cultivators that can control more weeds, can be used to replace labor inputs. If such alternatives are not offered, it is likely that domestic demand for these products will transfer to foreign suppliers who can provide them at a lower price due to their reduced labor costs, hence fostering a low-cost economy [34].

### 4.4. Mulching

Mulching is one of the cultural practices adopted to control or suppress weed growth in container-grown ornamentals. Mulches act as a physical barrier and suppress weed growth either preventing weed seeds from germination and emergence by light exclusion, by the release of allelopathic chemicals, or by acting as a physical barrier. In general, mulches can be both organic (shredded bark, residues of plants, hardwood chips, rice hull, etc.) and inorganic (plastic material, rocks, etc.) types [36]. Organic mulching substances are considered to be very effective in controlling annual small-seeded weeds. But, in the case of perennial weeds mulches tend to be less effective because of weeds' nature in incorporating a considerable amount of strength in their roots or due to the innate ability of their underground parts to overcome the strong layer of mulch with their respective shoots, once they start growing or germinating [37]. However, inorganic mulching materials are much more prevalent in landscaping and field crops [36].

According to Amoroso et al. [38], both mulching and chemical control have similar abilities to suppress weed growth, because the plants which were mulched with biodegradable discs produced more dry weight of shoots when compared to the plants that were non-treated and non-mulched.

A study conducted by Giaccone et al. [39] showed that biodegradable chitosan-based mulching spray can control weeds effectively in container production. Biodegradable

chitosan-based mulching spray is a derivative of chitin (chitin is second most available polysaccharide on the earth) composed of cationic carbohydrate biopolymer, which is generally insoluble in water but easily dissolves in most solutions of dissolved organic acids, such as acetic acid. In one study, mulching spray extracted from the scraps of crabs, shrimp and lobsters controlled the growth of weeds effectively in containers even under drastic infestation by weeds after its application, for not less than 2 months because of its film-forming nature. But it was also observed that mulch started degrading after 3 months of its application, which allowed a very small number of weeds to grow in containers. When compared with the performance of the herbicide oxadiazone, the biodegradable mulching spray showed better performance [39].

*4.5. Irrigation Management*

Container nursery growers are highly concerned with utilizing current water resources efficiently. To enhance water management, it is crucial to understand the current practices followed in commercial container production [40]. Nursery growers commonly prefer overhead irrigation as the most practical and regularly used irrigation system for the container production of woody ornamentals. However, drip or microjet irrigation are often used practically for materials grown in containers larger than 20 L [41]. An overhead irrigation system's infrastructure allows for great flexibility in terms of irrigating various container plants of variable sizes within an area. However, because container plants have a restricted root zone, they must be watered frequently, which can reduce the efficiency of irrigation in overhead sprinkler systems. The amount of water that is applied during irrigation and remains in the root zone so that plants can use it is referred to as the irrigation application efficiency. The irrigation system's infrastructure, the spacing of the container plants, the physical properties of the substrate, and the regularity of water distribution during irrigation are some of the elements that affect how well plants are watered or the efficiency of application [42].

According to Wilen et al. [22], subirrigation is an effective technique that could be employed in landscaping and certain farming methods to help reduce water loss due to evaporation while also potentially reducing weed growth. However, when using subirrigation for weed control, it is also crucial to be cautious and avoid excessive humidity of the soil or growing medium. For example, the study conducted by Wilen et al. [22] demonstrated that in subirrigated containers, the surface of the potting mix remained dry, preventing weed seeds from finding a suitable environment for germination and growth. Although different mulch depths had no effect on *Rhaphiolepis indica* L. growth as measured by dry weight, subirrigation had a negative impact on root, shoot, and total plant weight. Despite efforts to adjust irrigation times and frequency, the potting mix in the bottom half of subirrigated containers frequently became waterlogged. Because of this, most of the root growth occurred along the container wall and in the top half of the potting mix, compromising plant growth in the subirrigated treatments [22].

According to Stewart et al. [25], when compared to overhead systems, there is little knowledge about the impact of micro irrigation or drip irrigation techniques on weed management. Because only a portion of the substrate surface is moistened during each irrigation cycle, these methods are expected to reduce weed growth, especially in larger containers. This, however, may have unintended consequences for weed control, such as inhibiting the germination of some weed seeds, while potentially creating new problems, like how the use of micro or drip irrigation systems may cause issues such as ineffective herbicide activation due to scarcity of rainfall and might pose a risk of phytotoxicity due to insufficient overhead irrigation to remove herbicide residues from plant foliage, if any overhead applications are employed [25].

In context, there are plenty of options available for choosing the components, designs, and operation of irrigation systems to function in nurseries. Unfortunately, it is often noticed that the specific requirements of plants are generally neglected during the design phase of irrigation systems in nurseries. As the cost of water and water restrictions continue

to keep on rising, it will become increasingly important to consider these needs. To achieve optimal irrigation efficiency, it is crucial to properly design overhead and micro irrigation systems that provide consistent water delivery, based on the plant's demand and with an appropriate amount of irrigation water scheduled [42].

### 4.6. Substrate Stratification

The process of "substrate stratification" entails layering several substrates or the same substrate with various textures in nursery containers. It has been recently suggested that using this technique will improve drainage, control substrate moisture levels, and increase nutrient use effectiveness. In theory, a layer substrate made up of larger particle bark on top and smaller particle bark at the bottom of the container would allow for rapid drying of the surface, which would inhibit weed germination while also retaining adequate moisture for crop development [43]. A recent study conducted by Khamare et al. [43] showed that coarse bark (<1.27 cm or 1.9 cm particle size) when used as the top substrate and finer bark (<0.96 cm particle size) when used as the bottom substrate in the container can reduce the growth of the weed species bittercress (*Cardamine flexuosa* With.) by 80% to 97%, and the liverwort (*Marchantia polymorpha* L.) coverage decreased by 95% to 99%.

By stratifying the substrate, a more favorable gradient of air and water can also be created to promote the growth and establishment of plants grown in containers. The traditional industry method of filling containers uniformly with a similar substrate leads to the lower part of the container remaining at or near full saturation, while the upper part where the plant is located drains rapidly and has less water readily accessible. The demand for excessive irrigation during the first development and establishment stages of these systems may be reduced by modifying the hydraulic characteristics of the top layer of the growing substrate. This change would make it possible to maintain a rooted liner with a root ball that is one-third to half the depth of the container while consuming less water, reducing irrigation volume and leachate in the process [44].

According to Fields and Criscione [45], the use of peat in horticulture is being scrutinized as consumer knowledge of peat-related environmental sustainability concerns rises. The horticultural industry has been forced to search for peat substitutes as a result. Substrate stratification is one such option, which includes vertically layering many media in a single container. According to studies, this strategy can increase resource efficiency by using less water and fertilizer, especially in substrates used for nurseries. However, the results showed that it is possible to grow profitable greenhouse plants like petunia while also lowering peat usage by more than 50% in terms of volume by overlaying high-priced peat-based medium over inexpensive pine bark [45].

In addition, stratifying the substrate may provide weed management advantages akin to mulching. Furthermore, this approach may have an edge over standard mulch materials such as pine bark nuggets or rice hulls, which are typically applied to the top layer of nursery containers. Growers have suggested that this method can be implemented using their existing equipment, but a cost–benefit analysis is necessary to ascertain if the benefits of substrate stratification surpass the rise in labor costs [43].

### 4.7. Mulch Discs (or Geo Discs) and Lid Bags

A typical weed disc is shaped like a circle with a center aperture or slit that enables it to be wrapped around the plant's stem. The ideal features of a weed disc consist of being effortless to apply, resisting displacement due to wind, lying flat and fitting tightly on the container substrate while allowing water to pass through, preventing weed germination and growth on its surface, it should be obtainable in a numerous range of sizes, and it should be sturdy and economical. Although weed discs have desirable properties, such as the ability to prevent weed growth and germination, weeds can still emerge through the elongated slit or around the container's inner rim. While using two weed discs in offset positions can improve control, it also increases control expenses [46]. In certain cases, due

to excessive overhead irrigation, algae and mosses can start growing on top of plastic discs or geotextile discs, which can create additional problems.

A variety of products have been used or have the potential to be used in the Pacific Northwest (PNW) region of the United States along with Canada. Geotextile discs, coco discs, plastic discs, sawdust, Biotop, hazelnut (*Corylus avellana* L.) shells, and crumb rubber are among the materials used. Geotextile discs are a type of fabric made of polypropylene that is not woven and has one side coated with cupric hydroxide. Coco discs, on the other hand, are produced as a result of the processing of coconut (*Cocos nucifera* L.), in which longer fibers are extracted from the fruit pith of coconuts and used to make a variety of products, including weed discs. The thickness of coco disks is approximately 0.6 cm (0.25 in). Hazelnut shells are produced as a byproduct of the processing of hazelnut tree nuts and are crushed to a size of less than 0.6 cm (0.25 in). Plastic weed discs have been made in a variety of designs, but the majority are made of a thin and stiff plastic material that covers the surface of the container and includes preformed holes for water and air infiltration. Crumb rubber is made by removing the steel radials from tires and shredding the rubber components. Crumb rubber can be manufactured in a variety of sizes, all of which are less than 0.6 cm in length (0.25 in) [47]. In a study, Tex-R Geodisc (Texel USA, Henderson, NC, USA), a copper coated nonwoven polypropylene disc, was able to control weeds in containers for six months [48]. Based on this information, it appears that using weed discs properly in container nurseries could be a practical and cost-effective alternative to chemical weed control methods. When compared to herbicide-treated bark, the black polyethylene sleeve known as the Mori Weed Bag and plastic lids known as Enviro LIDs provided inferior control of weeds in containers. Nursery growers have explored Enviro LIDs, which are plastic lids with perforated holes for watering that are designed to be placed over the top of the container [2].

According to Chong [46], among the non-chemical weed management techniques explored for containers, Weed Guard, Tex-R Geodisc, Biodisc, and Enviro LIDs are the only ones currently available on the market and, unfortunately, there are few studies on the usefulness of Enviro LIDs in the literature in real time.

### 4.8. Fertilizer Placement

Strategic fertilizer placement might be a feasible approach to control weeds in certain ornamental crops in container production, given that these ornamental species could be harmed in herbicide application during weed control [49]. However, the outcome can be varied because of varying responses from different plant species, the source of fertilizers, the method of application, and the amount applied, especially with different types of growing media. Many research studies have manifested that it is essential to assess the source of fertilizer, fertilizer quantities, and the application techniques of fertilizers used in container production [50].

According to Fain et al. [51], the placement of fertilizer has the potential to influence weed seed germination by influencing the availability of nutrients required for their growth. This is because certain seeds require a sufficient supply of nutrients to germinate and thrive. When the fertilizer is dibbled, it reduces the quantity of nitrogen, phosphorus, and potassium present on or near the surface of the container where the weed seeds typically sprout. This can create multiple challenges for small-seeded weeds, such as prostrate spurge (*Euphorbia maculate*), which have limited nutrient stock and struggle to acquire the necessary nutrients they need in containers, where fertilizer has been dibbled [51]. However, according to Hickleton [52], it is generally not advised for growers to employ dibbling due to the probable root damage produced by excessive salt concentrations, which is based on the observation of relatively low root weights found in the dibbled fertilizer placement method [49].

Similarly, according to Altland et al. [53], weed establishment and their growth were reduced across various herbicide rates by dibbling fertilizer rather than topdressing or incorporating it. Weed control was excellent even at lower herbicide rates when Controlled-

Release Fertilizers (CRFs) were used, indicating that modifying fertilizer management could potentially result in lower herbicide rates. Also, crop shoot growth appeared to be similar when fertilizer was dibbled as compared with the topdressing of Controlled-Release Fertilizers (CRFs), and it exhibited slightly better growth than when incorporating the fertilizer into the soil [50].

However, according to Saha et al. [49], while previous research has focused on the effects of topdressing, incorporating, and dibbling fertilizer on weed growth, the impact of subdressing fertilizer on weed growth has not received much attention. So, if subdressing or dibbling were found to reduce weed growth or seed production significantly, nursery producers could use these alternative fertilizer placements as part of an integrated weed management program. This would aid in reducing the overall weed pressure in nursery crops grown in containers [49].

Ultimately, comprehending the impact of cultural practices such as fertilizer placement on weed control when utilizing regularly or frequently used herbicides can assist producers in efficiently managing their crops and weed control regimen [53].

At present, the nursery industry views the labor-intensive integrated weed management method as the sole practical strategy for weed control in potted plants. This method involves adhering to strict nursery hygiene standards while utilizing a combination of coir mats or bark mulch, herbicides, and physical hand weeding. Herbicides, mulch, and coir mat prices have all gone up in step with the rest of the supply chain. However, the effectiveness of the herbicides that are now available in the market is declining, and they may soon be subjected to regulations with rising environmental concerns [54].

## 5. Prospects and Future Directions

In a restricted growth environment, such as container plant production, weeds have been shown to reduce marketability and crop growth significantly. Most research focuses on weed competition in agronomic crops and limited research has been conducted on weed competition in container-grown ornamental plants.

More research needs to focus on this area as there are various types of ornamental plants with unique needs and many of them are very sensitive to herbicide injuries. Research data are required from various locations across the United States and even from different parts of the world where climatic and environmental conditions are varying. In addition, limited information is available on the effects of weed densities and container sizes on ornament–weed competition within containerized production and how the concepts of fertilizer placements (types and different depths) can be used efficiently to control weeds in containers without employing any herbicides on the ornamentals. Hence, there is an immediate need to evaluate the interference and competitive effects of pernicious weed species in container-grown ornamentals in the North Central United States and to develop effective non-chemical weed control strategies by altering fertilizer placement in container production.

**Author Contributions:** Conceptualization, D.S.; writing—original draft preparation, G.A.; writing—review and editing, D.S.; supervision, D.S.; funding acquisition, D.S. All authors have read and agreed to the published version of the manuscript.

**Funding:** This research was funded by the Michigan State University Project GREEEN (Generating Research and Extension to meet Economic and Environmental Needs), grant number GR22-014. This research was also supported by the United States Department of Agriculture (USDA) National Institute of Food and Agriculture, Hatch project number MICL02670.

**Data Availability Statement:** Not applicable.

**Conflicts of Interest:** The authors declare no conflicts of interest. The funders had no role in writing of the manuscript or in the decision to publish the results.

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
