# Peer review of "Emerging Perspectives on Non-Chemical Weed Management Tactics in Container Ornamental Production in the United States"

_horticulturae, doi:10.3390/horticulturae10030281_

Round 1
Reviewer 1 Report
Comments and Suggestions for Authors
The work titled "Emerging Perspectives on Non-Chemical Weed Management Tactics in Container Ornamental Production: An In-depth Review" summarizes some of the main alternative practices to the use of chemical tools for weed control in container-grown ornamental plants. As a first impression, I would like to raise a methodological concern. For a literature review, the selection of articles is crucial to support your conclusions. Papers for a good literature review should be up-to-date and have a significant impact on the topic. The articles in this review are, on average, old (many were written over 10 years ago) and often lack sufficient scientific impact (many are practical guides or proceedings). Another issue is the number of articles used: often in various paragraphs, only 4 papers are used to support your statements (e.g., paragraphs 4.1, 4.2, 4.3, 4.4). All these elements are in disagreement with what is stated in the title, where you claimed to have conducted an "in-depth review."
Additionally, I expect from a review, applicable information in a broader scientific context, while in this case, the work focuses on a narrow geographical area. The choice of a national journal instead of an international one like Horticulturae could be more appropriate for the publication of this manuscript. I find the part of paragraph 2 off-topic; there is too much economic information that does not contribute to the paper. If you want to emphasize the economic importance of the sector and the issue related to weed control, presenting a graph and briefly commenting on it would suffice. Lastly, use an international measurement system for presenting numerical data.
Based on these considerations, I think that the papers collected for this review are not of sufficient quality, not in adequate quantity, and they are generally quite old works. So far, the manuscript is more suitable for a technical report than for a scientific review in an international agronomy journal. The topic is nonetheless interesting, and I encourage the authors to improve the methodology for writing the review and resubmit it.
Comments on the Quality of English LanguageGood
Reviewer 2 Report
Comments and Suggestions for Authors
The review topic is interesting and necessary, and the structure proposed to address the issue seems to me to be appropriate. However, I believe that some parts should be updated in order to publish the article. It would also be necessary to expand the content in some parts providing some examples or updating information, in which only two or three references are cited, to improve the quality of the review. For example, some parts where the information needs to be improved or updated are the sections: 4.3. hand weeding (costs), 4.5. irrigation management, 4.7 mulch discs, 4.8 lid bags.
The references section also needs to be revised.
I propose a major revision.
Other suggestions about the text are detailed below.
Line 39: this reference is more recent for costs: Ingram, D. L., Hall, C. R., & Knight, J. (2017). Comparison of three production scenarios for Buxus microphylla var. japonica ‘Green Beauty’marketed in a No. 3 container on the west coast using life cycle assessment. HortScience, 52(3), 357-365. https://doi.org/10.21273/HORTSCI11596-16
Line 42: According to [4] the.. --> According to Knudson [4] the..
Line 60: air --> soil-air
Line 63: estimate of $500 --> estimate cost of $500
Line 157: it is more appropriate to write: can affect the growth of an ornamental crop. Case et al (2005) use this term.
Line 158: According to [13]... --> According to Berchielli-Robertson [13]...
Line 160: It would be interesting to give an example.
Line 162: According to [14].. --> According to Khamare et al. [14]..
Line 229: According to [22].. --> According to Diver et al. [22]..
Line 260: According to [27], to.. --> According to Wiles et al. [27], to..
Line 352: 4.0 --> 4
Line 355: (Gilliam et al., 1990) --> [16]
Line 355-358: This information does not refer to reference 32.
Line 377: According to [36], --> According to Amoroso et al. [36],
Line 381: A study conducted by [37], --> A study conducted by Giaccone et al. [37],
Line 395-396: It is said that it is necessary to know the current practices for commercial production. The reference is from 1992. If possible, it would be advisable to include a more recent reference.
Line 396-399: Same as before.
Line 409: According to [21], --> According to Wilen et al. [21],
Line 403: conducted by [21], --> conducted by Wilen et al. [21],
Line 422: According to [24], --> According to Stewart et al. [24],
Section "4.5. Irrigation Management" --> Almost no research results are provided for the control of most weeds by irrigation management.
Line 458: According to [43], --> According to Fields and Criscione [43],
Substrate Stratification Section: No example is given of any research that has studied how substrate stratification affects weed control. Only the hypotheses of various authors are indicated. It would be interesting to support this reasoning and cite an example.
Line 500-505: This reference is from 20 years ago. It would be interesting to comment on whether they are still marketed today. From this statement one can think that it is not a viable weed control method.
Line 512: According to [44], --> According to Chong [44],
Line 512-515: The same reference is cited in the sentence 3 times. Eliminate the repeated ones.
Lid Bags section: It would be interesting to add some more examples/references. There is little information.
Line 525: According to [48], --> According to Fain et al. [48],
Line 532-533: according to [49], --> according to Hicklenton [49],
Line 536: according to [50], --> according to Altland et al. [50],
Line 544: according to [46], --> according to Saha [46],
Line 562-563: In a restricted growth environment, such as container plant production, weeds have been shown to reduce marketability and crop growth by up to 60% [15] --> Perhaps it is drastic to make this statement literally, it would be necessary to qualify what happens in a specific experiment. It would be necessary to try to support this statement with other studies.
Line 565: [15], reported --> Fretz [15], reported
Line 568: [52], reported --> Wilbourn and Rauch [52], reported
Line 572: In 1989, [53], reported --> In 1989, Walker and Williams [53], reported
Line 574: Recently a study conducted by [14] --> Recently a study conducted by Khamare et al. [14]
References section.
It would be advisable to add the doi numbers in the references.
line 609: Link does not work.
line 616: complete the reference.
Line 627: use the same format for all references. Journal name without abbreviations.
Line 633: Weed Competition in Container Grown Japanese Holly1. --> The number 1 is not part of the title
Line 644: Complete the reference. ATTRA --> Appropriate Technology Transfer for RuralAreas
Line 645: Check the reference.
Line 650: Complete the reference.
Line 651-652: Complete the reference.
Line 657: Complete the reference.
Line 659: If this reference refers to this document : https://ucanr.edu/sites/sjcoeh/files/309014.pdf --> Replace the reference for "Wilen CA. 2018. UC IPM Pest Notes: Weed Management in Lanscapes. UC ANR Publication 7441. Okland, CA". The reference that appears in Scholar google is wrong.
Line 661: Complete the reference.
Round 2
Reviewer 1 Report
Comments and Suggestions for Authors
With the corrections from the first round, the paper has improved, but there are still some elements to be modified. Below are some specific comments.
Introduction: In general, the objectives of the review are not clear and should be specified. I would avoid using economic data as they are of little scientific relevance in the way they have been presented. I suggest focusing more on the environmental aspects associated with the extensive use of herbicides. In this context, there is a vast literature that has not been utilized to date.
Line 13: What do you mean by "competition for oxygen"?
Line 36: Why is "POST" written in uppercase letters?
Line 40: In general, I don't like when values are stated in terms of money. Costs and prices can vary significantly over time.
Line 40-45: It might be better to express this data as a percentage of the total plant cost; otherwise, it's unclear whether 0.0833 is a high or low amount.
Line 61-71: I find it quite unsuitable to report these economic data. We understand that weed management has a significant economic impact. Furthermore, data that you have reported are from a paper dated 2003 and may not be suitable for today. I would recommend focusing on the environmental impact of the extensive use of herbicides, as it is a much more relevant aspect.
Line 88: reference 8 is too old. What is the situation now?
Line 81-136: section: Importance of Ornamental Crop Production. In my opinion, this section is not relevant to the article's purpose. These values are specific to particular years and do not contribute to the understanding of the subsequent paragraphs.
Line 138-141: You have repeated the exact same sentence in the abstract, and this is not good.
Line 142: paper [13] do not mention about ornamental sector. Provide additional references.
Line 145-146: Numerous? they also mentioned that it depends on the weed species.
Line 145: In a scientific paper, units of measurement must be expressed in accordance with the International System of Units (SI).
Line 241: reference 26 does not provide insights regarding the fact that manual weeding can be ineffective
Line 309: Why did you use curly brackets?
Line 341-347: these are not important information from scientific point of view.
Line 438: same comment of line 145
Line 553-556: This is a conclusion, so you shouldn't include data and references in this section. You've presented a series of data, some of which are already mentioned in the text, and I don't see their utility. I would remove this part.
Reviewer 2 Report
Comments and Suggestions for Authors
Dear authors,
I appreciate your time and effort in reviewing the manuscript. I find the modifications made satisfactory, as well as the responses to my questions, which have been clear and helpful.
I value the interesting work you have carried out, and I believe it will contribute significantly to the advancement of knowledge in this area.
Author Response
We authors would like to thank Reviewer 2 for commenting that the modifications made in round 1 were satisfactory. Since no further revisions were suggested by Reviewer 2, we authors have no attachment/ response document for Reviewer 2 comment.